# Safety of Thread-Embedding Acupuncture: A Multicenter, Prospective, Observational Pilot Study

**DOI:** 10.3390/healthcare12232396

**Published:** 2024-11-29

**Authors:** Seojung Ha, Suji Lee, Bonhyuk Goo, Eunseok Kim, Ojin Kwon, Sang-Soo Nam, Joo-Hee Kim

**Affiliations:** 1Department of Acupuncture and Moxibustion Medicine, College of Korean Medicine, Sangji University, Wonju-si 26339, Republic of Korea; sangjiacu2@sangji.ac.kr; 2Department of Acupuncture and Moxibustion, Kyung Hee University Medical Center, Seoul 02447, Republic of Korea; sjstarry41@naver.com; 3Department of Acupuncture & Moxibustion, Kyung Hee University Hospital at Gangdong, Seoul 05278, Republic of Korea; goobh@khnmc.or.kr (B.G.); dangun1966@khu.ac.kr (S.-S.N.); 4Department of Acupuncture and Moxibustion Medicine, Pusan National University Korean Medicine Hospital, Yangsan 50612, Republic of Korea; eskim@pusan.ac.kr; 5KM Science Research Division, Korea Institute of Oriental Medicine, Daejeon 34054, Republic of Korea; cheda1334@kiom.re.kr; 6Department of Acupuncture & Moxibustion, Kyung Hee University College of Korean Medicine, Kyung Hee University Hospital at Gangdong, Seoul 05278, Republic of Korea; 7Research Institute of Korean Medicine, Sangji University, Wonju-si 26339, Republic of Korea

**Keywords:** thread-embedding acupuncture, polydioxanone, safety, prospective, observational study

## Abstract

**Background/Objectives**: Thread-embedding acupuncture (TEA) is widely used for cosmetic and therapeutic purposes; however, its safety profile, particularly in real-world clinical settings, remains under-researched. This study aimed to evaluate the safety profile of TEA through a prospective, observational analysis and confirm the feasibility of the study design for future studies involving larger patient populations. **Methods**: A multicenter, prospective observational study was conducted involving 100 patients who received TEA. Adverse events (AEs) were tracked, including incidence, severity, and duration during the 6-month post-treatment period. Bivariate analysis was used to assess factors influencing AE occurrence, including treatment site, depth, and patient-specific variables. **Results**: A total of 100 patients received 136 treatments during the study period. A total of 12 AEs were reported, most of which were mild and transient local reactions, including pain and bruising. More than half of the AEs occurred on the day of the procedure, with an average duration of 7 days. No serious AEs were observed, and all events resolved without any lasting effects. Patients undergoing multiple treatments showed no significantly higher AE rates than those receiving a single session. **Conclusions**: This study suggested that TEA generally has a favorable safety profile, with most AEs being mild and resolving without long-term effects. Further studies that evaluate the safety of TEA treatment across larger populations are recommended.

## 1. Introduction

Thread-embedding acupuncture (TEA) is a unique form of acupuncture involving the insertion of absorbable threads into specific acupoints to produce sustained therapeutic effects. It is frequently performed to treat a variety of conditions, including musculoskeletal disorders [1,2,3], facial palsy [4,5], obesity [6], and aging-related issues such as wrinkles [7]. TEA has two key advantages over traditional acupuncture. The inserted threads remain in the tissue for an extended time, providing continuous mechanical stimulation to the acupoints and surrounding areas. This sustained effect is particularly beneficial for managing chronic conditions [8,9]. TEA has evolved from traditional catgut-embedding therapy with the introduction of biodegradable materials such as polydioxanone, polycaprolactone, and polylactic acid-glycolic acid. These threads vary in their composition and degradation rates, contributing to different therapeutic outcomes [10,11,12,13]. As the threads dissolve, they stimulate surrounding connective tissue, enhancing collagen production and improving skin elasticity, which contributes to cosmetic benefits like skin tightening and wrinkle reduction [8,9].

Unlike traditional acupuncture, TEA is more invasive due to the insertion of absorbable materials into the body and the nature of the needle used, which resembles a Quincke-type needle that allows for thread insertion. This can cause greater tissue damage, leading to localized bleeding, swelling, and pain. In rare cases, localized allergic reactions, such as erythema or swelling, may occur due to the foreign body reaction triggered by polydioxanone threads [14]. Foreign body reaction involves an acute inflammatory phase followed by chronic inflammation, which, while beneficial for collagen synthesis, may sometimes lead to complications such as keloid or hypertrophic scar formation in susceptible individuals [12]. Adverse events (AEs) associated with TEA are influenced by both procedure-related factors, such as needle size, thread material, proper hygiene, and practitioner expertise, as well as patient-specific characteristics, including comorbidities like diabetes, autoimmune disorders, or connective tissue diseases. Combined with improper techniques or substandard materials, these factors can lead to complications such as infections, delayed healing, or chronic inflammation. Infections may occur if hygiene protocols are not strictly followed [15], and improper insertion depth can result in complications such as dermal hyperplasia or dimpling [16,17,18]. Specific patient conditions, such as diabetes, altered immune responses or impaired tissue regeneration, further increase the risk of these adverse outcomes [19]. Previous studies have reported AE rates of approximately 10% for local reactions (e.g., erythema, swelling) and 1% for systemic reactions (e.g., fever, systemic inflammation) [20]. Despite its widespread use, comprehensive safety evaluations of TEA remain limited, with most evidence coming from case reports and reviews [15,16,17,18,20,21], often lacking long-term follow-up.

This pilot study systematically investigated the safety profile of TEA in real-world clinical settings. Specifically, the type and frequency of AEs associated with TEA, their potential associations with patient-specific factors (e.g., comorbidities) and procedural variables (e.g., needle length, insertion depth), and the feasibility of conducting long-term safety monitoring of TEA were investigated. This study aims to address these gaps and inform future research.

## 2. Materials and Methods

### 2.1. Study Design and Setting

This was a preliminary prospective observational study designed to collect safety data on TEA, as it is frequently used in clinical Korean medicine practice. This study aimed to evaluate the safety of TEA by observing AEs in patients receiving TEA as part of their routine clinical care. This study was conducted at two university-affiliated hospitals: Kyung Hee University Korean Medicine Hospital at Gangdong and Pusan National University Korean Medicine Hospital. This study commenced on 1 September 2022, and concluded on 7 July 2023. Each participant was followed-up for a period of 6 months post-treatment.

This study included adults aged ≥ 19 years who received TEA at one of the participating hospitals during the study period. All participants received a detailed explanation of the clinical study and voluntarily provided written informed consent prior to inclusion. Patients with conditions or circumstances that the investigator determined could interfere with study participation, such as severe comorbidities or inability to comply with study protocols, were excluded. The trial was conducted in accordance with the Declaration of Helsinki and Good Clinical Practice guidelines. Ethical approval for this study was obtained from the institutional review boards of Kyung Hee University Korean Medicine Hospital at Gangdong (approval No.: KHNMCOH 2022-05-001, approval date: 9 May 2022) and Pusan National University Korean Medicine Hospital (approval No.: PNUKHIRB 2022-05-002, approval date: 17 June 2022). This study was registered with the Clinical Research Information Service (registration number: KCT0007494) on 30 June 2022.

### 2.2. Outcome Measures

The primary safety endpoint of this study was to assess the incidence of AEs within six months following TEA. Additionally, we aimed to evaluate the rate of treatment discontinuation due to AEs as a secondary outcome measure.

Safety data were collected through direct visits or self-reporting at predetermined intervals: baseline (immediately before the procedure), 2 weeks and 1, 3, and 6 months post-treatment. For the 2-week, 1-month, and 3-month follow-ups, the participants could self-report AEs using an application or diary without requiring an in-person visit. The final follow-up at six months required an in-person visit for a comprehensive evaluation. During these follow-ups, we documented various types of data related to AEs, including incidence rates, types, and patterns of AEs, and their distributions across different age and gender groups. We also recorded the outcomes of these events and the specific interventions performed to manage them. We also recorded the outcomes of these events and the specific interventions performed to manage them. AEs were classified into local and systemic adverse effects, and their severity was categorized as mild, moderate, or severe. In addition, interactions between TEA and pre-existing conditions or concurrent medications were documented. When AEs occurred, the WHO causality assessment was conducted to evaluate the likelihood that the TEA procedure caused the adverse event [22]. The results of this assessment were categorized as “Probable”, “Unlikely”, or “Possible”, based on the temporal relationship and clinical evidence. Detailed information regarding the onset, resolution, duration, and actions taken to manage these AEs was recorded.

### 2.3. Procedures and Enrollment

Participants aged ≥ 19 years who underwent TEA at one of the participating hospitals were recruited for this study. The TEA procedures were performed by licensed practitioners with over 10 years of clinical experience. The practitioners were given the flexibility to tailor the treatment according to the patients’ symptoms, in alignment with their clinical expertise and the treatment environment. This allowed the practitioners to apply TEA techniques that were most appropriate for the patient’s condition. Detailed information about each TEA procedure was recorded, including the target disease, TEA products used (including manufacturers), types of threads (composition and form), treatment areas, number of thread insertions, thread sizes (gauge), needle lengths and thicknesses, and depth of thread insertion. For participants undergoing multiple TEA treatments, additional monitoring for AEs was performed 2 weeks after each procedure.

Detailed information regarding each TEA procedure was collected, including the target disease, TEA products used (including manufacturers), types of threads (composition and form), treatment areas, number of thread insertions, thread sizes (gauge), needle lengths and thicknesses, and depth of thread insertion.

For follow-up monitoring, the participants self-reported AEs via an application or diary at 2 weeks, 1 month, and 3 months post-treatment. The final follow-up at 6 months required an in-person visit for a comprehensive evaluation. For participants undergoing multiple TEA treatments, AEs were additionally monitored 2 weeks after each procedure. During each follow-up, the participants were asked about local symptoms, such as pain, foreign body sensation, bleeding, bruising/hematoma, pigmentation changes, redness, swelling, suppuration, dimpling, tremor, stiffness, nodules/granulomas, thread visibility/exposure, sensory changes, itching, nerve damage, and salivary gland injury. They were also queried about systemic symptoms including fever, chills, fatigue, myalgia, infection, dizziness, shock, and angioedema. For each reported symptom, details such as the onset date, resolution date, duration, related tests, actions taken to address the symptom, and recovery status were recorded.

### 2.4. Data Sources and Measurement

Demographic information and medical history were recorded on pre-designed case report forms. Medical history data included any diagnoses made within the past three years, concurrent conditions, history of adverse reactions to TEA, medications taken within the last three months, and all medications taken after enrollment in the study. For participants undergoing an additional TEA procedure, information on AEs that occurred after the previous procedure was documented in the case report forms. The severity of AEs, interventions required to manage AEs, outcomes of AEs, and time to complete recovery were recorded in detail for all participants.

### 2.5. Sample Size

As a preliminary study, we aimed to assess the feasibility of conducting a clinical trial, including the appropriate timeframe, data collection items, and methods of collection, such as patient self-reporting via a mobile application, and to investigate the basic information for the evaluation of TEA safety in clinical practice. The sample size of 100 participants was pragmatically determined based on practical constraints, such as study duration, recruitment feasibility, and available resources. This approach aligns with the recommendations outlined by Arain et al., which emphasize that the sample size in feasibility studies should be determined based on the study’s objectives and practical considerations rather than rigid numerical thresholds [23].

### 2.6. Statistical Analysis

All statistical analyses were performed using the SAS^®^ Version 9.4 (SAS Institute. Inc., Cary, NC, USA) software. Statistical analyses were performed to evaluate the safety data collected in this study. Continuous variables were summarized using means and standard deviation, whereas categorical variables were presented as frequencies and percentages. Bivariate analysis was performed to investigate the association between various factors and the occurrence of AEs after TEA procedure. The factors analyzed included gender, previous treatment experience, history of AEs from previous treatments, presence of comorbidities, number of procedures (1 vs. ≥ 2), treatment site, number of insertions, thread length, thickness, and depth. Fisher’s exact test and odds ratio was used for categorical variables. Statistical significance was set at *p* < 0.05.

## 3. Results

### 3.1. Baseline Characteristics

A total of 102 participants were screened from the unselected population for this study, and two were dropped for violating the inclusion criteria; thus, a total of 100 participants were enrolled. All enrolled participants completed six months of follow-up, with no dropouts. The safety endpoint, defined as the incidence of adverse effects, revealed that 12 AEs were reported in 10 patients. All AEs were mild and resolved without any long-term sequelae. No serious AEs were observed.

The baseline characteristics of the participants, including gender, age, height, weight, comorbidities, vital signs, target disease, previous TEA experience, and history of adverse effects, are presented in Table 1.

### 3.2. Adverse Events Related to TEA

The incidence of AEs after the TEA procedure is summarized in Table 2. Twelve AEs were reported in 10 patients. All the AEs were mild and transient local reactions. The most frequently reported AEs included pain (six cases) and bruising (five cases). Common complications typically associated with TEA treatment, such as skin depression, contour irregularities, visible threads, thread extrusion, infection, nerve damage, and vascular damage, were not observed during the study period. No medical intervention was required for any case, except for one patient who was managed with an ice pack, considered a simple and non-invasive measure. All patients recovered fully, and 12 cases resolved without any long-term sequelae. When considering the number of TEA sessions, three reported AEs occurred in patients who underwent only one TEA session, whereas seven AEs were reported in patients who underwent two or more sessions. The average time for the onset of AEs was 21 days post-procedure, and the average duration of AEs was 8.5 days. No serious AEs occurred during the study period.

The timing of AEs after the first procedure showed that 58.3% of the events (seven cases) occurred within the first 14 days. No additional AEs were observed after 3 months, highlighting the early concentration of events. This pattern is also depicted in Figure 1. The cumulative incidence of AEs following TEA was analyzed over a 6-month period. More than half of the AEs occurred within the first 14 days, with the highest incidence occurring on the day of the procedure. Seven out of ten patients reported localized pain or bruising on the day of the procedure. One patient with ankle pain experienced treatment site pain on day 12 after the first TEA procedure and foreign body sensation with tension on day 11 following the second procedure. Another patient with headache reported treatment site pain on days 23 and 83 after the first procedure. One patient with lower back pain experienced stabbing pain in the ischium area 80 days post-procedure, which was unrelated to the thread-embedding site and likely aggravated by external factors, such as studying for exams. This case was categorized as “Unlikely” based on the WHO causality criteria, whereas all other cases were classified as “Probable”.

The duration of the AEs varied, with an average duration of 16.3 days for all symptoms. Outliers were noted, particularly for pain, which lasted up to 99 days. These outliers, including one case categorized as ‘Unlikely’ based on the WHO causality assessment criteria, were excluded from the analysis. Excluding the outliers, the average duration of pain was 7.3 days, and the overall average duration was 8.8 days.

The dashed line indicates the 14-day mark, highlighting the early concentration of adverse events, with 58.3% (7 out of 12 adverse events) occurring within this period. Adverse events occurring beyond 14 days were rare, with only one event at 81 days categorized as ‘unlikely’ based on the WHO causality criteria. All other events were classified as ‘probable’.

### 3.3. Bivariate Analysis

Bivariate analysis was conducted to explore factors associated with the occurrence of AEs following TEA procedure, as shown in Table 3. The analysis found no significant associations between gender, comorbidities, previous TEA experiences, or treatment site and the occurrence of AEs (all *p*-values > 0.05). However, the number of TEA sessions was significantly associated with AEs, with patients undergoing more than one session showing a higher likelihood of experiencing AEs (*p* = 0.0078, OR = 0.63).

Further analysis of other factors, such as needle length, thickness, insertion depth, and manufacturer, revealed no significant differences in the occurrence of AEs. This suggests that while the number of TEA sessions was a significant factor in the occurrence of AEs, other factors like treatment site and needle specifications did not show a meaningful impact on the incidence of adverse events.

Additionally, the number of threads inserted was considered a potential factor for the occurrence of AEs. Therefore, the analysis focused on the facial and lumbar sites, where the most treatments were performed. However, no significant associations were found for the facial (*p* = 0.0532) and lumbar (*p* = 0.9999) sites. Since all types of thread used were polydioxanone, a separate bivariate analysis was not conducted for this variable.

## 4. Discussion

This study prospectively investigated the incidence, timing, duration, and characteristics of AEs following 6 months of TEA in a real clinical setting with 100 patients, providing an updated safety profile. The incidence of AEs was low (8.82%), all of which were mild and recovered without sequelae. Similarly, a systematic review of adverse events [20], including 45 randomized controlled trials and 16 case reports, found that local reactions such as bleeding, erythema, swelling, and pain occurred in 9.84% of cases. In contrast with our study, systemic reactions were reported in about 1% of cases, and five serious AEs, such as necrosis and ulceration, were observed. These serious AEs were linked to the use of catgut, which has been associated with a higher risk of foreign body reactions and inflammation [20]. Since our study exclusively used polydioxanone, no systemic reactions or serious AEs were observed, suggesting that polydioxanone may contribute to a safer profile compared to catgut.

In a survey of Korean medicine doctors [24], bruising and pain were the most common AEs that did not require medical intervention, while dimpling was the most frequent AE that required treatment. In our study, dimpling did not occur, which is likely because we primarily used 29 G mono-type threads to treat diseases, especially facial paralysis. For facial paralysis, 29 G mono-type threads of 30–40 mm in length are commonly used [24], which may explain the specific AEs observed in this study. On the other hand, according to the survey, 67% of the respondents used TEA for cosmetic purposes. TEA for cosmetic purposes is often similar to thread lifting, which utilizes cog-type threads with thicker needles and threads compared to mono-type threads. These thicker needles and threads are more likely to cause adverse effects such as inflammation and infection, and dimpling is a common complication of the lifting procedure [25]. Although longer and thicker needles are generally expected to cause more tissue damage, our study did not find a significant correlation between needle length or thickness and the occurrence of AEs. The use of 29 G, 30–40 mm mono-type threads was more commonly associated with AEs, but the use of longer needles (60–90 mm) and thicker 27 G needles accounted for less than 10% of the total TEA procedures, with none being associated with AEs in the AE group. This suggests that the higher incidence of AEs in the 30–40 mm, 29 G group could be attributed to factors such as treatment site or patient-specific characteristics rather than needle specifications. However, future studies would benefit from recruiting more patients who undergo treatment with longer and thicker needles, which may help to better understand the potential impact of needle specifications on the occurrence of adverse events. In line with the survey, factors such as insertion depth and patient-specific characteristics were considered essential contributors to AE occurrence. However, our study did not find significant correlations between treatment site or patient characteristics and AEs. While myofascial treatments showed a trend toward lower AE incidence, statistical significance was not reached. Interestingly, the number of TEA procedures performed was associated with AE occurrence, with patients undergoing a single procedure exhibiting a lower AE incidence compared to those who underwent multiple procedures. Since polydioxanone typically resorbs over 3–6 months, it is plausible that the frequency of TEA procedures within a specific time frame may influence the occurrence of AEs [12].

In this study, 58.3% of all AEs occurred within 14 days. In particular, seven AEs occurred on the day of the procedure, including two cases of pain and five cases of bruising. Pain on the day of the procedure and for several hours afterward is a common symptom. This pain is not considered a complication, as it is a natural local tissue response to the procedure. While all reactions were reported as AEs, it may be more appropriate to consider pain lasting longer than five days as an AE, as mentioned in the review study in [20], rather than categorizing short-term pain as a natural procedure reaction. Most AEs resolved within about 9 days, but future studies should also consider evaluating pain severity using a 10-point scale. Most AEs occurred within 14 days, but since some have been reported to occur rarely up to three months post-procedure, future studies should focus on closely monitoring AEs within the first two weeks and conducting long-term evaluations for at least three months.

One limitation of this study was the lack of blinding, which may have introduced bias in the reporting and evaluation of AEs. However, this risk is generally overestimated in open-label studies, and the use of a patient-reported application and diary minimized recall bias, as patients were able to report AEs in real time [26]. Additionally, the combination of physician observations and patient-reported outcomes allowed for a more comprehensive analysis, incorporating both subjective symptoms and objective clinical findings. Therefore, the safety results of TEA treatment in this study are more reliable. Future multicenter studies should consider including local clinics, not just university hospitals, to better reflect real-world clinical practice.

## 5. Conclusions

This prospective study provides valuable preliminary data on the safety profile of TEA in a real clinical setting. The incidence of AEs was low, with only local reactions observed and no serious or systemic reactions. Most AEs resolved quickly, but given the higher initial occurrence rate, monitoring within the first two weeks and follow-up for up to three months are recommended. While the frequency of TEA procedures was associated with AE occurrence, no clear correlation was found with other factors. Future multicenter studies, including local clinics, are essential to better reflect real-world clinical practice and further validate these results.

## Figures and Tables

**Figure 1 healthcare-12-02396-f001:**
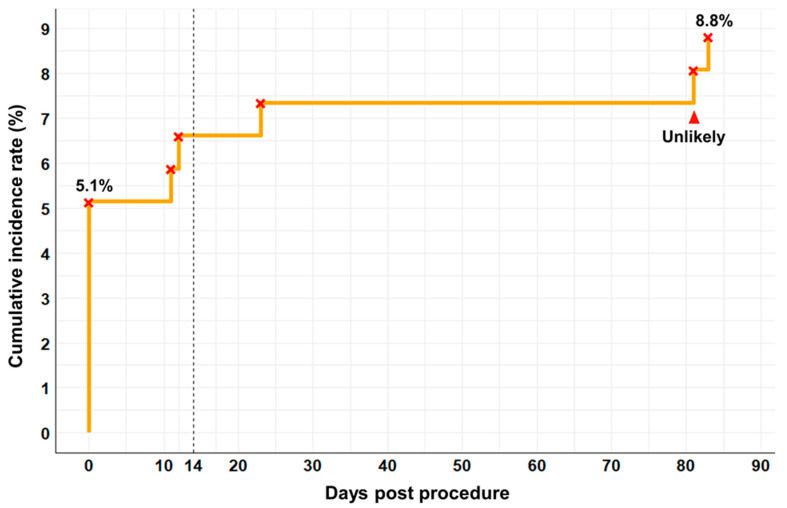
Cumulative incidence of adverse events following thread-embedding acupuncture procedure.

**Table 1 healthcare-12-02396-t001:** Baseline characteristics of the study participants.

	Total	Adverse Events	No Adverse Events
Number of participants	100	10	90
Gender (Male/Female)	32 (32.0%)/68 (68.0%)	2 (20.0%)/8 (80.0%)	31 (33.3%)/61 (66.7%)
Age (year)	49.1 ± 14.5	47.5 ± 11.8	49.3 ± 14.8
Height (cm)	163.1 ± 8.3	159.9 ± 11.0	163.5 ± 7.9
Weight (kg)	64.5 ± 14.1	59.4 ± 10.2	65.0 ± 14.4
Participants with comorbidities	38 (38.0%)	3 (30.0%)	35 (38.9%)
Type of comorbidities			
Cardiovascular disease	12 (12.0%)	1 (10.0%)	11 (12.2%)
Musculoskeletal disorders	16 (16.0%)	2 (20.0%)	14 (15.6%)
Urogenital disorders	4 (4.0%)	0 (0.0%)	4 (4.4%)
Endocrine disorders	14 (14.0%)	0 (0.0%)	14 (15.6%)
Digestive disorders	2 (2.0%)	0 (0.0%)	2 (2.2%)
Other conditions	6 (6.0%)	1 (10.0%)	5 (5.6%)
Total	54 (54.0%)	4 (40.0%)	50 (55.3%)
Vital sign			
Systolic blood pressure (mm/Hg)	124.5 ± 15.1	125.8 ± 14.1	124.3 ± 15.2
Diastolic blood pressure (mm/Hg)	74.7 ± 9.5	75.7 ± 13.4	74.6 ± 9.1
Pulse rate (bpm)	74.6 ± 10.4	78.6 ± 17.5	74.2 ± 9.3
Temperature (°C)	36.5 ± 0.2	36.4 ± 0.1	36.5 ± 0.2
Conditions treated with TEA			
Cosmetic purposes	1 (1.0%)	0 (0.00%)	1 (1.1%)
Low back pain	27 (27.0%)	1 (10.0%)	26 (28.9%)
Neck pain	15 (15.0%)	0 (0.0%)	15 (16.7%)
Shoulder pain	8 (8.0%)	0 (0.0%)	8 (8.9%)
Elbow pain	1 (1.0%)	0 (0.0%)	1 (1.1%)
Ankle pain	3 (3.0%)	1 (10.0%)	2 (2.2%)
Facial paralysis	56 (56.0%)	7 (70.0%)	49 (54.4%)
Other conditions	2 (2.0%)	1 (10.0%)	1 (1.1%)
Previous TEA experiences			
Yes	72 (72.0%)	7 (70.0%)	65 (72.2%)
No	28 (28.0%)	3 (30.0%)	25 (27.8%)
History of TEA-related adverse events			
Yes	1 (1.0%)	0 (0.0%)	1 (1.1%)
No	99 (99.0%)	10 (100.0%)	89 (98.9%)

Data are presented as n (%) for categorical variables and mean ± SD for continuous variables. TEA, Thread-embedding acupuncture.

**Table 2 healthcare-12-02396-t002:** Adverse events reported in study participants.

Characteristics	Value
Number of patients	100
Patients with one or more adverse event	10 (10.0%)
Number of TEA session	
1 session	3 (30.0%)
More than 1 session	7 (70.0%)
Total Visit	136
Number of adverse events	12 (8.8%)
Treatment-related adverse events *	
Pain	5 (45.5%)
Bruise	5 (45.5%)
Foreign body sensation and tension	1 (9.1%)
Severity of adverse events	
Mild	12 (100.0%)
Moderate	0 (0.0%)
Severe	0 (0.0%)
Management of adverse events	
TEA maintained	12 (100.0%)
TEA stopped	0 (0.0%)
Timing of adverse events after first-procedure	
Within 14 days	7 (58.3%)
14 days–1 month	3 (25.0%)
1 month–3 months	2 (16.7%)
Over 3 months	0 (0.0%)
Timing of adverse events after last procedure	
Within 14 days	8 (66.7%)
14 days–1 month	2 (16.7%)
1 month–3 months	2 (16.7%)
Over 3 months	0 (0.0%)
Duration of adverse events †	
Pain	7.3 ± 4.5
Bruise	8.8 ± 3.4
Total	8.8 ± 4.3
Outcome of adverse events	
Recovery without sequelae	12 (100.0%)

Data are presented as n (%) for categorical variables and mean ± SD for continuous variables. TEA, Thread-embedding acupuncture. * One case of pain was excluded as it was categorized as “Unlikely” based on the WHO causality assessment criteria. † Outliers were excluded from the analysis to calculate the duration of adverse events.

**Table 3 healthcare-12-02396-t003:** Occurrence of adverse events following the thread-embedding acupuncture procedure.

	Total (N = 136)	Adverse Events (N = 11)	No Adverse Events (N = 125)	*p*-Value	Odds Ratio (95% CI)
Gender (Male/Female)	40/96	2/9	38/87	0.5072	0.51 (0.10, 2.47)
Comorbidities (Yes/No)	53/83	4/7	49/76	0.9999	0.89 (0.25, 3.19)
Previous TEA experiences (Yes/No)	93/43	8/3	85/40	0.9999	1.25 (0.32, 4.98)
Number of TEA session					
1 session/More than 1 sessions	100/36	4/7	96/29	0.0078 *	0.17 (0.05, 0.63) †
TEA treatment site					
Facial	89 (65.4%)	8 (72.7%)	81 (64.8%)	0.7476	1.45 (0.37, 5.74)
Neck	16 (11.8%)	0 (0.0%)	16 (12.8%)	0.3609	-
Shoulder	11 (8.1%)	0 (0.0%)	11 (8.8%)	0.5993	-
Lumbar	29 (21.3%)	1 (9.1%)	28 (22.4%)	0.4564	0.35 (0.04, 2.82)
Buttocks	5 (3.7%)	0 (0.0%)	5 (4.0%)	0.9999	-
Upper limb	1 (0.7%)	0 (0.0%)	1 (0.8%)	0.9999	-
Lower limb	10 (7.4%)	2 (18.2%)	8 (6.4%)	0.1870	3.25 (0.60, 17.64)
Suture size					
6–0	128 (95.5%)	11 (100.0%)	117 (95.1%)	0.9999	-
5–0	9 (6.7%)	0 (0.00%)	9 (7.3%)	0.9999	-
Needle length					
25 mm	22 (16.2%)	0 (0.0%)	22 (17.6%)	0.2109	-
30 mm	3 (2.2%)	1 (9.1%)	2 (1.6%)	0.2251	6.15 (0.51, 73.84)
38 mm	71 (52.2%)	7 (63.6%)	64 (51.2%)	0.5362	1.67 (0.46, 5.98)
40 mm	34 (25.0%)	4 (36.4%)	30 (24.0%)	0.4666	1.81 (0.50, 6.61)
60 mm	11 (8.1%)	0 (0.0%)	11 (8.8%)	0.5993	-
90 mm	1 (0.7%)	0 (0.0%)	1 (0.8%)	0.9999	-
Needle thickness					
30 G	22 (16.2%)	0 (0.0%)	22 (17.6%)	0.2109	-
29 G	106 (77.9%)	11 (100.0%)	95 (76.0%)	0.1217	-
27 G	11 (8.1%)	0 (0.0%)	11 (8.8%)	0.5993	-
Insertion depth					
Subcutaneous	33 (24.3%)	2 (18.2%)	30 (24.8%)	0.9999	0.67 (0.14, 3.29)
Myofascial	130 (95.6%)	9 (81.8%)	121 (96.8%)	0.0748	0.15 (0.02, 0.93) †
Intramuscular	43 (36.4%)	4 (36.4%)	39 (31.2%)	0.7419	1.26 (0.35, 4.56)
Manufacturer of TEA					
Dongbang	43 (31.6%)	4 (36.4%)	39 (31.2%)	0.7419	1.26 (0.35, 4.56)
Hyundae Meditech	93 (68.4%)	7 (63.6%)	86 (68.8%)	0.7419	0.79 (0.22, 2.87)

Data are presented as n (%). TEA, thread-embedding acupuncture; G, gauge. * *p*-value < 0.05 indicates a statistically significant difference. † Odds ratio is statistically significant.

## Data Availability

The original contributions presented in this study are included in this article. Further inquiries regarding additional data can be directed to the corresponding author.

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
