# Peer review of "Safety of Thread-Embedding Acupuncture: A Multicenter, Prospective, Observational Pilot Study"

_healthcare, 2024, doi:10.3390/healthcare12232396_

Round 1
Reviewer 1 Report
Comments and Suggestions for Authors
This is a readable, interesting and useful short paper. I hope the following comments will be useful to the authors (numbers refer to the line numbers in the paper).
22. This is not really a 'large' study at all - the authors themselves (270) suggest that 'larger patient populations' should be investigated in future research.
63. 'will document' - or just 'documents'?
138. How was required sample size estimated?
144. SPSS: Version and date> Company and its location?
146. Authors should justify use of means and CIs. Were any statistical tests conducted to justify this choice? In Table 2 they also use medians and IQRs. Why?
152. I would prefer 'a t-test' rather than just 't-test'. Again, were any statistical tests conducted on the data to justify using this method?
175. Isn't an ice-pack 'specific'?
179. Here the use of 'average' time to onset is confusing: 21 days, and yet more than half occurred within 14 days! (with 7 out of 10 occurring on the day of the procedure - 225). Figure 1, for me, only adds to the confusion. Pain (or other AE) distribution is not clear from the Figure.
188-9. 'Immediate', and yet AEs occurred on Days 80, 83. I would argue that this indicates that continuing close monitoring is therefore essential - On the basis of what is written, I do not agree with the authors' statement (253-4) that 'future studies should shorten the total observation period and focus on the acute phase within the first week of treatment'. They will have to argue this more convincingly.
269. It might therefore be better to leave out 'short-term' here.
245. ‘Risk of bias‘ generally overestimated in open-label studies – support this statement with a reference or other evidence?
Tables:
1. Explain AEY, AEN, %, whether numbers for Age, Height and Weight are min-max ranges or IQRs. What does the double '+' sign indicate?
2. Means (95% CIs) and Medians (IQRs) are both included, but it is unclear whether both CIs and IQRs are indicated by parentheses, or if the latter are shown using squaee brackets. What does the '+' sign indicate?
3. Percentages would be clearer if following in line with the frequencies!
Explain ‘Standard and ‘G’ (gauge?), the term ‘subcuticular’, and use of '*'.
References
Capitalising and abbreviation of words in Journal titles are not uniformly consistent; likewise for words in article titles.
Translations and English should be checked - e.g., 325: 'an' should be used instead of 'a'.
Supplementary Material
The material included is not appropriate. On the other hand, 288, the data availability statement suggests that information on this can be found in the Supplementary Material. It is not.
Reviewer 2 Report
Comments and Suggestions for Authors
The topic is very relevant, since thread-embedding acupuncture is a little-studied treatment method. Moreover, this method is widely used throughout the world and its prevalence is likely to increase in the coming years.
The title matches the content.
Add from the literature data on the development of allergic reactions to the material used or cases of the development of colloidal scars. Is there a connection between AEs and concomitant diseases?
Can you add information about the advantages of this method over classical acupuncture?
The purpose should be more specific.
The same information is repeated several times:
· In lines 93-94: Safety data were collected through direct visits or self-reporting at predetermined intervals: baseline (immediately before the procedure), 2 weeks and 1, 3, and 6 months.
· In lines 110-113: Follow-up monitoring was structured as follows: at two weeks, one month, and three months post-treatment, participants could self-report AEs using an application or diary without the need for an in-person visit. The final follow-up at six months required an inperson visit for a comprehensive evaluation.
Please indicate “Procedures and Enrollment” point by point. It's not clear.
What method was used to determine the sample size?
Change please data analysis to statistical analysis.
Add additional statistical information about the methods used for normality testing, to assess the equality of variances between two or more groups, to test the differences between the means of the groups, to prevent data from incorrectly appearing to be statistically significant and to compare the means of the same variable between groups.
Add, please, inclusion and exclusion criteria.
In lines 159-160: “All enrolled participants completed six months of follow-up, with no dropouts. The safety endpoint, defined as the incidence of adverse effects, revealed that 12 AEs were reported in 10 patients.” It's not clear. Indicate, please, how many complications were identified in each of the 10 patients.
There are many unclear abbreviations in the table 1: HEY AEN, SBP, DBP, Pulse.
Please indicate in the table description which numbers are in brackets:
Age: 49.11 (46.27, 51.95) 47.50 (39.04, 55.96) 49.28 (46.22, 52.34)
Height: (cm) 163.1 (161.4, 164.8) 159.9 (152.0, 167.7) 163.5 (161.8, 165.1)
Weight: (kg) 64.49 (61.68, 67.30) 59.41 (52.13, 66.69) 65.04 (62.01, 68.07)
SBP: 124.5 (121.6, 127.4) 125.8 (115.7, 135.9) 124.3 (121.2, 127.5)
DBP: 74.74 (72.88, 76.60) 75.70 (66.11, 85.29) 74.63 (72.76, 76.50)
PULSE: 74.61 (72.57, 76.64) 78.60 (66.09, 91.11) 74.17 (72.24, 76.11)
Temp: 36.51 (36.47, 36.54) 36.44 (36.36, 36.52) 36.51 (36.48, 36.55)
In line 172: “All the AEs were mild and transient local reactions.” How about 2 patients with complications lasting from 1 to 3 months (listed in Table 2 and figure 1).
The title of column 2 in table 2 is “Frequency (percent) or Mean (95% CI)/Median (IQR)”. Although the values ​​listed below are absolute and percentage. Please clarify.
Pain during the procedures and for several hours after the procedure is not a complication. This is a natural local tissue response to damage. In addition, subjective pain is more painful and it would be interesting if the severity of this pain syndrome was determined on a 10-point scale.
In table 2. Please explain and show in the text how the percentages between the brackets were obtained. What numbers were the absolute numbers divided by?
I would remove figure 1. It's confusing. Late complications are recorded in 22-28, 86-88 days with a complication rate of 10%!!!
Indicate in Table 3 what length, depth, thickness mean?
In table 3. In the columns: total, AEY and AEN and below “therapy site”. What are the values indicated after fraction sign?
What I could understand from Table 3: Reported complications occurred after procedures with lengths of 36–40 mm rather than 60–90 mm and with 29 G rather than 27 and 30 mm needles. Please explain in the text.
In discussion, in lines 213-214: A survey of Korean medical doctors revealed that the most commonly reported AE that resolved without medical intervention was bruising, whereas the most common AE that required treatment was dimpling” Add reference.
In line 215: “Most of these AEs, including bruising, pain, and foreign body sensation, resolved within 4 weeks.” Add reference.
In lines 218-219: “A literature review of 45 RCTs and 16 case reports identified 28 types of AEs. The most common AEs were local reactions such as bleeding, redness, swelling, and pain, with systemic reactions reported in approximately 1% of cases.” Add freference.
In lines: 224-225: I don't think 8.8% is low for AE. It is indicated just above (In lines 219-220) that AEs reported by other authors occurred in only approximately 1% of patients.
Paragraph between lines 251-255: This study found that most AEs occurred on the day of the procedure and lasted for approximately 7 days, with no significant safety concerns during the 6-month follow-up period. Based on these findings, future studies should shorten the total observation period and focus on the acute phase within the first week of treatment. Additionally, future multicenter studies should consider including local clinics, not just university hospitals, to better reflect real-world clinical practice.” Move it to the beginning of discussion.
Paragraph between lines 258-262: “This study, conducted in a real-world clinical setting with one hundred patients, confirmed the safety of TEA. The observed incidence and types of AEs were consistent with those reported in previous studies, further supporting the notion that TEA is a generally safe procedure when performed by trained professionals. These findings provide valuable data for the growing body of evidence regarding TEA and offer a reliable basis for its continued use in clinical practice” You have already indicated this above in other words. In my opinion, it would be better if you removed it.
In conclusion< lines 265-266: “with AE rates similar to those previously reported, and no serious AEs.” What reports? In the discussion you mentioned two: one with 1%, the second with AE 21.5%.
Overall the discussion is weak. First, you need to indicate your results. Next, compare with other works and indicate how these results differ from each other and it is necessary to additionally compare the side effects of this technique with similar ones, mainly acupuncture and thread lifting.
It is necessary to indicate the main complications that may occur during this procedure: skin dimpling, contour irregularity, visible threads, and thread extrusion, infection, hyperpigmentation, nerve and vascular damage, foreign body sensation.
Additionally, it is unclear what complications have been reported in 22-28, 86-88 days.
Conclusion needs to be redone.
Round 2
Reviewer 2 Report
Comments and Suggestions for Authors
Thank you for giving me the opportunity to review this manuscript again.
As I said before the topic is very relevant due to the lack of clinical researches devoted to the safety of the use of thread-embedding acupuncture in treatment of many diseases.
Many successful changes have been made to the introduction. More information on allergic reactions associated with polydioxanone threads, which are linked to foreign body reactions involving acute and chronic inflammation were added . Moreover, authors addressed the potential for keloid and hypertrophic scar formation as complications in susceptible individuals. The connection between AEs and patient-specific factors, such as diabetes and altered immune responses, has also been highlighted. In addition, a detailed explanation of the benefits of TEA over classical acupuncture is included. Added interesting information regarding the cosmetic benefits of TEA such as collagen production and increased skin elasticity.
The title of the article matches the content.
The purpose of the research has been shortened and specified.
Many changes were made in materials and methods that truly improved the quality and scientific rationale of the manuscript.
Research methods were described in detail with appropriate explanations.
More statistical information about the methods used for normality testing, to assess the equality of variances between two or more groups, to test the differences between the means of the groups, to prevent data from incorrectly appearing to be statistically significant and to compare the means of the same variable between groups has been added to statistical analysis section.
The results are written very concisely and very briefly. Many changes have been made to the tables, figures and text, which are presented point by point. This significantly improved the quality of the manuscript and increased its scientific originality, soundness and validity.
In my opinion, the purpose of the study is fully realized.
Discussion and conclusions follow logically from the results of the study and are fully consistent with the purpose of the study.
Conclusions is directly related to the data that was collected and analyzed.